# Impact of Pharmacist-Led Multidisciplinary Team to Attain Targeted Vancomycin Area under the Curved Monitoring in a Tertiary Care Center in Thailand

**DOI:** 10.3390/antibiotics12020374

**Published:** 2023-02-11

**Authors:** Kittiya Jantarathaneewat, Tuangrat Phodha, Kankanit Singhasenee, Panipak Katawethiwong, Nuntra Suwantarat, Bernard Camins, Thanawat Wongphan, Sasinuch Rutjanawech, Anucha Apisarnthanarak

**Affiliations:** 1Center of Excellence in Pharmacy Practice and Management Research, Faculty of Pharmacy, Thammasat University, Pathum Thani 12120, Thailand; 2Research Group in Infectious Diseases Epidemiology and Prevention, Faculty of Medicine, Thammasat University, Pathum Thani 12120, Thailand; 3Pharmacy Unit, Thammasat University Hospital, Pathum Thani 12120, Thailand; 4Paholpolpayuhasena Hospital, Kanchanaburi 71000, Thailand; 5Division of Infectious Diseases, Faculty of Medicine, Thammasat University, Pathum Thani 12120, Thailand; 6Chulabhorn International College of Medicine, Thammasat University, Pathum Thani 12120, Thailand; 7Division of Infectious Diseases, Department of Medicine, Icahn School of Medicine at Mount Sinai, New York, NY 10029-6574, USA; 8Trat Provincial Public Health Office, Trat 23000, Thailand

**Keywords:** vancomycin, therapeutic drug monitoring, area under the curve, pharmacist-led intervention

## Abstract

Vancomycin Area Under the Curve (AUC) monitoring has been recommended to ensure successful clinical outcomes and minimize the risk of nephrotoxicity, rather than traditional trough concentration. However, vancomycin AUC monitoring by a pharmacist-led multidisciplinary team (PMT) has not been well established in Southeast Asia. This study was conducted at Thammasat University Hospital. Adult patients aged ≥ 18 years who were admitted and received intravenous vancomycin ≥48 h were included. The pre-PMT period (April 2020–September 2020) was defined as a period using traditional trough concentration, while the post-PMT period (October 2020–March 2021) was defined as a period using PMT to monitor vancomycin AUC. The primary outcome was the rate of achievement of the therapeutic target of an AUC/MIC ratio of 400–600. There was a significantly higher rate of achievement of therapeutic target vancomycin AUC during post-PMT period (66.7% vs. 34.3%, *p* < 0.001). Furthermore, there was a significant improvement in the clinical cure rate (92.4% vs. 69.5%, *p* < 0.001) and reduction in 30-day ID mortality (2.9% vs. 12.4%, *p* = 0.017) during the post-PMT period. Our study demonstrates that PMT was effective to help attain a targeted vancomycin AUC, improve the clinical cure rate, and reduce 30-day ID mortality. This intervention should be encouraged to be implemented in Southeast Asia.

## 1. Introduction

Vancomycin is a commonly used antibiotic for methicillin-resistant *Staphyloccocus aureus* (MRSA) infections as well as other gram-positive bacteria such as *Enterococcus faecium, Corynebacterium* spp., coagulase-negative *Staphylococci*, and *Streptococcus* spp. [1,2]. A well-known adverse effect of vancomycin is nephrotoxicity, which can be minimized by monitoring serum concentrations to avoid supratherapeutic levels [3]. Furthermore, the targeted pharmacodynamic index for MRSA infection is the steady-state 24-h area under the concentration-time curve (AUC24) divided by the minimal inhibitory concentration (AUC24/MIC). An AUC 24/MIC value above 400 is associated with higher clinical response [4], while the daily vancomycin Area Under the Curve (AUC) values (assuming a MIC of 1 mg/L) should not exceed 600 mg × hr/L to reduce the risk for nephrotoxicity [3,5,6]. In 2019, traditional vancomycin trough concentration monitoring, which aims to maintain trough concentrations between 15 and 20 mg/L for serious MRSA infection, was recommended as a surrogate marker for AUC [7]. This strategy was widely incorporated into routine practice, particularly in resource-limited settings. However, the traditional trough monitoring had limited clinical data support as an accurate prediction for AUC monitoring. Furthermore, trough concentration cannot be used as a guide for MRSA treatment if *S. aureus* MIC ≥ 2 [3,8]. Thus, achieving target vancomycin serum trough concentration may increase the risk for nephrotoxicity compared with AUC monitoring, especially trough concentration higher than 15 mg/L and AUC above 600 mg × hr/L [9,10].

In 2020, the guidelines on vancomycin monitoring were revised by the American Society of Health-System Pharmacists (ASHP), the Infectious Diseases Society of America (IDSA), the Society of Infectious Diseases Pharmacists (SIDP), and the Pediatric Infectious Diseases Society (PIDS). The guidelines recommended AUC/MIC monitoring over traditional trough concentration-guided dosing [3]. The recommended daily AUC range (assuming a MIC of 1 mg/L) was between 400–600 mg × hr/L [3]. The AUC can be estimated by using Bayesian software PrecisePK^®^ (PrecisePK version 1.5.1.1.5.1, Healthware Inc., San Diego, CA, USA) or calculated through two-point sampling. The trapezoid rule was used to estimate the vancomycin AUC in the past. However, this strategy is rarely used in routine practice because of the complexity of the calculation and the necessity of collecting several serum concentrations over the same dosing interval [3]. A first-order pharmacokinetic analytic equation can be used as an alternative to computing a vancomycin AUC from two-point sampling, as modified by Pai and Rodvold [11]. Despite the fact that this first-order pharmacokinetic analytic equation was simply and ease of use to estimate the vancomycin AUC in the routine practice, this strategy had some limitations to provide accurate vancomycin AUC when patient had acute kidney injury during or after sampling time and patient who received multiple dosage regimen within 24 h [3]. Furthermore, this strategy should be applied only in a steady-state condition, which might not be feasible for critically ill patients. However, this strategy demonstrated slightly over-predicted vancomycin AUC [11]. Therefore, the clinician should be applied with caution. The Bayesian software calculation was proposed as the recommended strategy for vancomycin AUC monitoring. This approach can be used within the first 24 h rather than at steady state, which can be applied to achieve rapid target pharmacokinetics/pharmacodynamics of vancomycin in some serious conditions, such as in critically ill patients. Even though Bayesian software calculation requires training, it is the gold standard for vancomycin AUC monitoring and can provide dosing recommendation in real time [3]. There are several commercial Bayesian software available, such as PrecisePK^®^, BestDose^®^, and InsightRx^®^. According to a previously published study, PrecisePK^®^ and BestDose^®^ had the most accurate estimates [12]. Unfortunately, the AUC-based dosing protocol is not widely used in Southeast Asia due to the prescribing culture, lack of knowledge on vancomycin therapeutic dose monitoring, lack of time to implement AUC monitoring, and poor communication and coordination of therapeutic drug monitoring processes [13,14,15]. In Thailand, there are a few studies that implemented vancomycin AUC monitoring by using Bayesian software [16,17]. One of the previous studies reported that a multidisciplinary team can improve adherence to a vancomycin AUC monitoring protocol [14]. In this study, we aim to compare whether pharmacist-led AUC monitoring versus traditional trough concentration monitoring can help attain target pharmacodynamics of vancomycin in a lower-middle income countries in Southeast Asia.

## 2. Materials and Methods

This retrospective study was conducted at Thammasat University Hospital, a tertiary care and teaching hospital, between April 2020 and March 2021. Adult inpatients (age ≥ 18 years old) who received intravenous vancomycin at least 48 h and had at least 1 vancomycin blood concentration were included. Exclusion criteria included patients who had chronic kidney disease stage 4 or higher, had an acute kidney injury, or were receiving renal replacement therapy. A pharmacist-led multidisciplinary team (PMT) was introduced in the hospital in October 2020. The pre-PMT period (April 2020–September 2020) was described as a period using traditional trough concentration-guided dosing, while the post-PMT period (October 2020–March 2021) was described as a period using PMT with vancomycin AUC monitoring. The primary outcome of this study was to compare the achievement rate of therapeutic targets identified by an AUC/MIC ratio of 400–600 between the pre- and post-PMT periods. The secondary outcomes were the clinical cure rate, nephrotoxicity events, length of stay, 30-day infectious disease mortality, vancomycin consumption as defined by the defined daily dose (DDD) per 1000 patient-days, and time to achievement of the therapeutic target. Data collected included patient baseline characteristics (e.g., gender, age, body mass index, comorbidities, Charlson comorbidities index), baseline serum creatinine, protein plasma level, sites of infection, causative pathogens, pharmacokinetic parameters (e.g., volume of distribution, vancomycin clearance, elimination rate constant, and half-life), serum vancomycin concentration, clinical cure, and nephrotoxicity events from chart review. A nephrotoxicity event was defined according to the Acute Kidney Injury Network (AKIN) and Kidney Disease: Improving Global Outcomes (KDIGO) clinical practice guidelines as follows: (1) increase in serum creatinine by ≥0.3 mg/dL within 48 h, (2) increase in serum creatinine to ≥1.5 times baseline within the prior 7 days, or (3) urine output volume <0.5 mL/kg/hr for 6 h [18]. Clinical cure was defined as the resolution of symptoms [19].

Our pharmacist-led multidisciplinary team consists of clinical pharmacists, infectious disease physicians, and laboratory technicians. The hospital’s vancomycin protocol, which included weight-based loading and maintenance dosing and appropriate timing of vancomycin peak and trough concentration collections within 72 h after vancomycin initiation, was developed based on current guidance and distributed hospital-wide (Appendix A) [3]. Traditional trough concentration monitoring based on 2009 guidelines from the American Society of Health-System Pharmacists (ASHP), the Infectious Diseases Society of America (IDSA), and the Society of Infectious Diseases Pharmacists (SIDP) was applied in routine practice during the pre-PMT period [7]. During the pre-PMT period, the clinical pharmacist’s therapeutic drug monitoring service was only performed when the primary physician notified the pharmacy unit. During the post-PMT period, the hospital’s revised vancomycin protocol was implemented (Appendix A) and the clinical pharmacists reviewed the vancomycin dosage after the first dose was prescribed and notified the infectious diseases physician for monitoring. Our recommended vancomycin dosage regimen was a loading dose of 20–30 mg/kg followed by a maintenance dose of 15–20 mg/kg or adjusted by renal function (Appendix A). The recommended vancomycin infusion rate did not exceed 10 mg/min. Serum vancomycin analyses were performed by VITROS^®^ 4600 (Ortho Clinical Diagnostics, Rochester, NY, USA). This method was based on immunoassay technology [20]. After the patient’s vancomycin serum concentration was reported, the clinical pharmacist performed the calculation using the online Bayesian software PrecisePK^®^ (PrecisePK version 1.5.1.1.5.1, Healthware Inc., San Diego, CA, USA) and reported the results to the infectious diseases physician. The PrecisePK^®^ software analyzes pharmacokinetic parameters based on individual variables such as sex, renal function, age, body weight, race, and critically ill status. The infectious diseases physician contacted the primary physician for any dose adjustment or the need for additional follow-up vancomycin concentration. A daily discussion on an appropriate dosage regimen was made by the PMT members via LINE^®^ (LINE application, LINE Corporation, Tokyo, Japan). In this study, we used the E-test method (M.I.C. Evaluator strip; Thermo Fisher Scientific, Basingstoke, UK) to determine vancomycin susceptibility and MIC for causative pathogens such as *S. aureus, E. faecium,* and coagulase-negative staphylococci. The MIC and susceptibility results for the causative pathogen in this study were interpreted based on Clinical and Laboratory Standards Institute (CLSI) guidelines 2020 [21]. In this study, we assumed a MIC of 1 mg/L for the participant who had no pathogens identified.

To achieve 80% power and a 95% confidence interval, the minimum sample size required in each period was 101 subjects. All statistical analyses were performed using STATA version 17 (College Station, TX, USA). The chi-square test (two-tailed) was used to compare proportions for categorical variables. An independent t-test was used to compare the means of continuous variables. Antibiotic consumption was reported as defined daily dose (DDD) per 1000 patient-days and calculated by using the usage quantity divided by the DDD conversion factor, which was indicated by the World Health Organization Collaborating Centre for Drug Statistics Methodology and the Norwegian Institute of Public Health [22]. Univariate and multivariate analysis of variables influencing on 30-day infectious diseases-related mortality were performed. All comparisons were 2-sided and a statistically significant was consider when *p* value <0.05. This study was approved by the Ethics Committee No.3 of Thammasat University (protocol code 115/2563).

## 3. Results

Overall, 210 patients were included. The proportion of male participants was similar in both groups. The mean age was 59 ± 19.4 years old. The most common comorbidities were hypertension, dyslipidemia, diabetes mellitus, malignancy, and chronic kidney disease (Table 1). The median Charlson comorbidity index was 3 (interquartile range (IQR) 2–5). The median creatinine clearance calculated based on the Cockcroft and Gault formula is 76 (IQR 52.5–98) ml/min. The mean plasma protein level was 2.8 ± 0.6 g/dL. The most common indications for vancomycin empirical therapy were skin and soft tissue infection (24.9%), bacteremia (16.2%), and respiratory tract infection (15.7%). The most common causative pathogens were *Enterococcus faecium* (9%) and *Corynebacterium* spp. (9%). Methicillin-resistant *S.aureus* (MRSA) and Methicillin-resistant coagulase-negative *Staphylococci* (CoNS) were found in approximately 10 of 210 patients (4.8%) and 5 of 210 patients (2.4%), respectively. The median vancomycin MIC for MRSA at our institution was 1 (IQR 1–1) mg/L (*p* = 0.134). There was no pathogen identified in 31.9% of the participants. The pharmacokinetic parameters were similar in both groups. There was no difference in pharmacokinetic parameters between genders except for vancomycin clearance. The male participants showed a higher vancomycin clearance than female participants (4.3 ± 2.3 vs. 3.7 ± 1.9, *p* = 0.027). The baseline characteristics are described in Table 1.

Compared with the pre-pharmacist-led multidisciplinary team (PMT) period, a significantly higher achievement of therapeutic target vancomycin AUC was shown during the post-PMT period (66.7% vs. 34.3%, *p* < 0.001). The mean supratherapeutic AUC in the pre-PMT period was higher than in the post-PMT period (638 ± 179.7 vs. 568.9 ± 178.5, *p* = 0.006) while the mean trough concentrations of both groups were within the range of 15–20 mg/mL (Table 2). The distribution of patients who had achieved optimal AUC and vancomycin trough concentrations is shown in Figure 1. The predicted AUC was similar to the AUC calculated after dose adjustment. Vancomycin trough concentrations between 10 and 14.9 mg/L in the post-PMT period had a higher proportion of patients who achieved the optimal AUC target (39.1% vs. 12.4%, *p* < 0.001). A significant improvement in clinical cure (92.4% vs. 69.5%, *p* < 0.001) and reduction in 30-day infectious disease mortality were found during the post-PMT period when compared with the pre-PMT period (2.9% vs. 12.4%, *p* = 0.017). Furthermore, the proportion of patients who achieved their targeted vancomycin level within 48 h was higher in the post-PMT period (*p* = 0.005), while vancomycin consumption (DDD per 1000 patient-day), nephrotoxicity events, and length of stay were similar in both periods. A subgroup analysis of primary and secondary outcomes for patients with hypertension, dyslipidemia, diabetes mellitus, male gender, and bacteremia and infections of unknown origin yielded similar results to the overall analysis (Appendix A). In the multivariate analysis, the post-PMT period showed a reduction in 30-day infectious diseases-related mortality (OR 0.22; 95%CI 0.06–0.84, *p* = 0.027) while being male (OR 4.51; 95%CI 1.21–16.84, *p* = 0.025) and having bacteremia (OR 3.78; 95%CI 1.07–13.39, *p* = 0.039) were associated with increased 30-day infectious diseases mortality (Table 3).

## 4. Discussion

There are some important findings in our study. First, a pharmacist-led multidisciplinary team is feasible and can help attain vancomycin target pharmacodynamics in lower-middle income countries in Southeast Asia. Second, such a strategy can help reduce 30-day infectious disease-related mortality and improve clinical cure rates. Third, AUC monitoring can help reduce the proportion of supratherapeutic vancomycin levels and achieve vancomycin therapeutic target levels faster. Lastly, vancomycin levels can be monitored in real time through easily accessed AUC monitoring using Bayesian software (PrecisePK^®^).

A pre- and post-implementation study conducted in the United States found that vancomycin AUC-based monitoring using the trapezoidal rule calculation from two-point sampling had achieved a higher proportion of therapeutic values during the post-implementation period (73.5% vs. 55%, *p* = 0.001). Notably, the proportion of supratherapeutic trough concentration was smaller in the post-implementation period (1.7% vs. 18%, *p* < 0.001) [23]. Likewise, Neely et al., found that vancomycin AUC monitoring using a Bayesian software (BestDose^®^) can better achieve therapeutic targets compared with trough concentration monitoring (70% vs. 19%, *p* < 0.001) [24]. A randomized controlled trial from Canada using Bayesian software (PrecisePK^®^) found that vancomycin AUC-based monitoring increased target attainment when compared with trough concentration monitoring (risk ratio 1.32, 95% confidence interval (CI) 1.01–1.32, *p* = 0.038) [25]. Similarly, the application of vancomycin AUC monitoring was associated with higher clinical cure rates when compared with trough concentration monitoring and was associated with a reduction in 30-day infectious disease-related mortality [14,26,27]. Similar to previous studies, our study showed the benefit of vancomycin AUC monitoring through a reduction in 30-day infectious disease-related mortality in lower-middle-income country in Southeast Asia. Consistent with previous studies, male and bacteremia were associated with 30-day infectious disease-related mortality in our study [14,26]. This is the first study to demonstrate the integration of a pharmacist-led multidisciplinary team to improve vancomycin AUC monitoring.

Vancomycin AUC monitoring has been associated with a reduction in nephrotoxicity in previous studies [5,6,17,24,25,26,28,29,30]. Notably, the incidence of nephrotoxicity events in our study was comparable to previous studies [23,27]. The lower nephrotoxicity events in our study might be because patients with chronic kidney disease (stage ≥ 4), acute kidney injury, or patients on renal replacement therapy were excluded. Consistent with previous studies, the proportion of patients with supratherapeutic vancomycin levels was significantly lower during the post-PMT period (18% vs. 7.1% *p* = 0.001) [23]. A previous study showed that vancomycin AUC monitoring can help achieve targeted vancomycin AUC faster than trough concentration monitoring (47.6% vs. 42.1%) [25]. The achievement of targeted vancomycin AUC during the first 2 days of treatment was associated with a declining rate of treatment failure [31]. Our study confirmed that vancomycin AUC monitoring can reduce the time to achievement of therapeutic targeted vancomycin AUC. The proportion of patients who achieved the therapeutic range within 48 h was higher in the post-PMT period (33.3% vs. 17.1%, *p* = 0.005), while the median time to achieve target was 3 days (*p* = 0.40).

Even though vancomycin was predominantly used as a first line to treat a serious methicillin-resistant *S. aureus*, vancomycin can be used to treat other gram-positive pathogens such as *Enterococcus* spp. [1]. The most common pathogen in our study was *E. faecium. The* Methicillin-resistant *S.aureus* (MRSA) and Methicillin-resistant coagulase-negative *Staphylococci* (CoNS) prevalences in this study were low when compared with previous studies [23,24,27,28]. Several literatures showed that the vancomycin AUC monitoring was associated with clinical outcomes including clinical cure, microbiological cure, and nephrotoxicity events for MRSA infections [3,4,5,23,24,27,28,29]. However, there are a few reports on vancomycin monitoring and clinical outcomes for enterococcal infections [1,32,33]. Although vancomycin AUC monitoring was recommended for MRSA infection, this strategy was controversial for enterococcal infection [1,32,33]. A retrospective study conducted in Singapore found that a vancomycin AUC/MIC value of ≥389 achieved within 72 h was associated with a 30-day mortality reduction in patients who had enterococcal bacteremia (Odd ratio 6.38; 95% CI 1.51 to 30.84, *p* = 0.01) [1]. Furthermore, a retrospective study in Thailand demonstrated that a vancomycin AUC/MIC of ≥400 can be used as an optimal target for enterococcal infections. The authors found that a vancomycin AUC/MIC of ≥400 showed less clinical failure (adjusted hazard ratio (HR) 0.50; 95% CI 0.26 to 0.97, *p* = 0.042) and less microbiological failure (adjusted HR 0.37; 95% CI 0.14 to 0.94, *p* = 0.036). However, a vancomycin AUC/MIC value of ≥400 was associated with a higher rate of nephrotoxicity, which may be affected by several factors such as concomitant nephrotoxic agents and different acute kidney injury definitions [32]. On the contrary, a retrospective study in South Korea found that vancomycin trough concentrations of 15 mg/L or more were associated with a lower 28-day mortality rate in patients who were infected with *Enterococcus* spp. (survivors 58.6% vs. non-survivors 12.5%, *p* = 0.042). However, they did not find an association between AUC/MIC and mortality outcomes (survivors 82.8% vs. non-survivors 75%, *p* = 0.479) [33]. Furthermore, Nakakura et al. found that there was no significant association between the vancomycin AUC/MIC or trough concentration and the 30-day all-cause mortality [34]. In spite of the fact that some participants in our study were predominantly infected with *Enterococcus* spp. (9%), there was no pathogen identified in most of our participants (31.9%). Even though our study demonstrated the effectiveness of vancomycin AUC monitoring, the effectiveness of this approach for enterococcal infection cannot be concluded due to an inadequate sample size. Thus, a larger prospective study of vancomycin AUC monitoring for enterococcal infection should be conducted.

In Japan and Taiwan, implementation of vancomycin AUC monitoring has led to a reduction in both nephrotoxicity events and 30-day mortality [26,30,35]. In Japan, there is Bayesian software that has been modified to target pharmacokinetic parameters specifically for the Japanese population [22,36]. Implementation of such a program resulted in a wider accessibility of vancomycin AUC monitoring and an improvement in the accuracy of vancomycin AUC monitoring. In Southeast Asia, there are only a few studies that implemented vancomycin AUC monitoring using Bayesian software [14,16,17]. Barriers to the implementation of vancomycin AUC monitoring in this region include a lack of knowledge about vancomycin therapeutic dose monitoring and a lack of good communication in the healthcare worker team (e.g., primary care physicians, infectious diseases physicians, nurses, and pharmacists) who are involved in vancomycin administration [14]. In Thailand, successful implementation of vancomycin AUC monitoring using Bayesian software was demonstrated in the critically ill population [16,17]. A previous study in Thailand showed that a higher AUC/MIC value (≥679) was associated with acute kidney injury events (*p* = 0.041). However, a higher AUC/MIC value at days 1 and 2 was not associated with treatment failure, 30-day mortality, or microbiological failure [17]. Additional research is needed to expand the role of vancomycin AUC monitoring in the non-critically ill population. To our knowledge, this is the first study in Southeast Asia that implemented hospital wide vancomycin AUC monitoring, featuring a pharmacist-led multidisciplinary team.

Our study has several limitations. First, this study is not a randomized controlled trial, which may have led to selection bias and the confounding factors which were not fully accounted for. Second, the lower incidence of nephrotoxicity in our study may have occurred because we excluded populations with kidney dysfunction. Further studies to fully evaluate nephrotoxicity outcomes may show a higher proportion of nephrotoxicity events. Third, the relatively short follow-up duration in our study may prevent us from evaluating the long-term outcomes of vancomycin AUC monitoring, such as hospital cost reduction, vancomycin consumption, and impact on vancomycin-resistant organisms. Fourth, because the number of confirmed infections with organisms susceptible to vancomycin was low, patients with infections susceptible to vancomycin would therefore be missed and would have been excluded from the analyses. In addition, some patients who do not have infections susceptible to vancomycin could have been inadvertently included. Finally, the relatively high cost of Bayesian software may limit the implementation of vancomycin AUC monitoring in many resource-limited settings in Southeast Asian countries. A cost-effectiveness study on vancomycin AUC monitoring may provide better insight on the proper implementation strategy of vancomycin AUC monitoring in Southeast Asia.

In conclusion, our study showed that AUC monitoring using Bayesian software (PrecisePK^®^) can be implemented in lower-middle income countries in Southeast Asia using a pharmacist-led multidisciplinary team, and it has been shown to be associated with a reduction in 30-day infectious diseases-related mortality and an improvement in clinical cure. Such a strategy can lead to faster achievement in vancomycin therapeutic target levels and is associated with a lower risk for supratherapeutic levels.

## Figures and Tables

**Figure 1 antibiotics-12-00374-f001:**
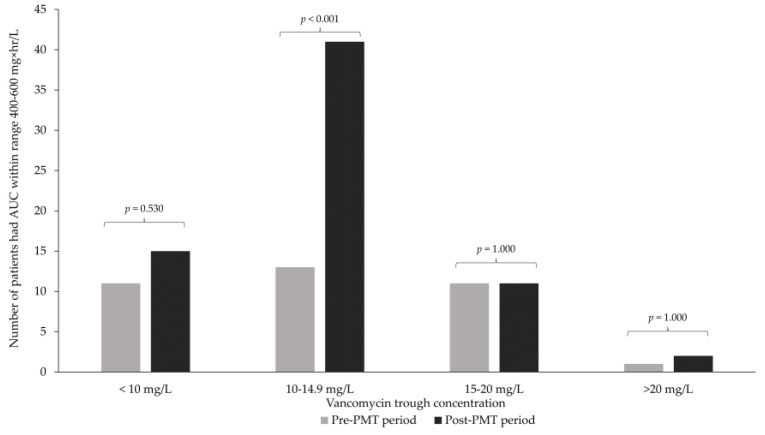
Distribution of the number of patients who achieved optimal vancomycin AUC and trough concentration.

**Table 1 antibiotics-12-00374-t001:** Baseline characteristics.

Characteristics	Total (*n* = 210) No (%)	Pre-PMT Period (*n* = 105) No (%)	Post-PMT Period (*n* = 105) No (%)	*p*-Value
Male	107 (51)	56 (53.3)	51 (48.6)	0.490
Age, mean years ± SD	59 ± 19.4	59 ± 20.7	59.1 ± 18.1	0.969
Body Mass Index, mean Kg/M^2^ ± SD	23.2 ± 5.5	22.6 ± 5.2	23.8 ± 5.7	0.108
Comorbidities				
Hypertension	100 (47.6)	47 (44.8)	53 (50.5)	0.407
Dyslipidemia	63 (30)	27 (25.7)	36 (34.3)	0.175
Diabetes mellitus	54 (25.7)	26 (24.8)	28 (26.7)	0.752
Chronic kidney disease	13 (6.2)	4 (3.8)	9 (8.6)	0.251
Charlson comorbidities index, median (IQR)	3 (2–5)	3 (2–5)	3 (1–5)	0.464
Serum creatinine, mean mg/dL ± SD	0.82 ± 0.35	0.82 ± 0.35	0.81 ± 0.34	0.902
Creatinine clearance, median mL/min (IQR)	76 (52.5–98)	70.3 (51.3–93.2)	80.3 (55.2–99.1)	0.118
Protein plasma level, mean g/dL ± SD	2.8 ± 0.6	2.8 ± 0.6	2.9 ± 0.6	0.480
Volume of distribution, mean L ± SD	48.1 ± 19.4	45.8 ± 19	50.4 ± 19.7	0.086
Vancomycin clearance, mean L/hr ± SD	4.0 ± 2.2	3.8 ± 2.3	4.2 ± 2	0.175
Elimination constant rate, mean 1/hr	0.09 ± 0.05	0.09 ± 0.06	0.09 ± 0.04	0.935
Half-life, mean hr ± SD	10.9 ± 9	11.8 ± 11	10 ± 6.5	0.148
Indication				
Skin and soft tissue infection	51 (24.9)	22 (21)	29 (27.6)	0.260
Bacteremia	34 (16.2)	13 (12.4)	21 (20)	0.134
Respiratory tract infections	33 (15.7)	21 (20)	12 (11.4)	0.088

IQR, interquartile range; SD, standard deviation.

**Table 2 antibiotics-12-00374-t002:** Study outcomes.

Study Outcomes	Total (*n* = 210) No (%)	Pre-PMT Period (*n* = 105) No (%)	Post-PMT Period (*n* = 105) No (%)	*p*-Value
Achieved therapeutic target	106 (50.5)	36 (34.3)	70 (66.7)	<0.001
Calculated AUC, mean mg × hr/L ± SD	603.4 ± 182	638 ± 179.7	568.9 ± 178.5	0.006
Predicted AUC, mean mg × hr/L ± SD	615.6 ± 227.9	668.5 ± 257.4	562.7 ± 227.9	0.007
Calculated AUC range				
AUC less than 400 mg × hr/L	17 (8.1)	9 (8.8)	8 (8.6)	1.000
AUC within range (400–600 mg × hr/L)	105 (50)	36 (34.3)	69 (56.7)	<0.001
AUC more than 600 mg × hr/L	88 (41.9)	60 (57.1)	28 (26.7)	<0.001
Trough concentration, mean mg/L ± SD	15.5 ± 6.3	16.6 ± 6.5	14.4 ± 5.9	0.011
Clinical cure	170 (81)	73 (69.5)	97 (92.4)	<0.001
30-days infectious diseases mortality	16 (7.6)	13 (12.4)	3 (2.9)	0.017
Nephrotoxicity event	7 (3.3)	3 (2.9)	4 (3.8)	1.000
Proportion of patient who achieved therapeutic range within 48 h	53 (25.2)	18 (17.1)	35 (33.3)	0.005
Time to target, median days (IQR) *	3 (2–5)	3 (2–5.5)	3 (1–5)	0.398
Vancomycin consumption, median DDD per 1000 patient-day (IQR)	8.5 (5.3–14.5)	8 (4.8–13.3)	9.5 (5.5)	0.095
Length of stay, median days (IQR)	31 (20–50)	31 (20–47)	33 (20–51)	0.436

AUC, Area Under the Curve; DDD, defined daily dose; IQR, interquartile range; SD, standard deviation, * total 48 patients in pre-PMT period and 78 patients in post-PMT period were reported.

**Table 3 antibiotics-12-00374-t003:** Multivariate analysis on 30-day infectious diseases mortality.

	Univariate Analysis			Multivariate Analysis		
	OR	95% CI	*p*-Value	OR	95% CI	*p*-Value
Post-PMT period	0.21	0.06–0.75	0.017	0.22	0.06–0.84	0.027
Bacteremia	2.59	0.84–7.99	0.099	3.78	1.07–13.39	0.039
Male	4.61	1.27–16.69	0.020	4.51	1.21–16.84	0.025

## Data Availability

The data that support the findings of this study are available on request from the corresponding author. The data are not publicly available due to privacy or ethical restrictions.

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
