# Peer review of "Impact of Pharmacist-Led Multidisciplinary Team to Attain Targeted Vancomycin Area under the Curved Monitoring in a Tertiary Care Center in Thailand"

_antibiotics, 2023, doi:10.3390/antibiotics12020374_

Round 1
Reviewer 1 Report
The manuscript is interesting. However, some issues must be clarified:
1. In my opinion FigureS1 should be in the main text.
2. Why did Authors not consider to divide the patients into subgroups (with hypertension, dyslipidemia, etc.)?
3. Did Authors cosider sex as a variable that may cause the differences in the pharmacokinetic analysis?
4. Why did Authors did not take into consideration the protein plasma levels?
Author Response
Response to Reviewer 1 Comments
Point 1: In my opinion FigureS1 should be in the main text.
Response 1: We added Figure S1 into the main text as suggested in page 6 and revised the wording for “figure S1” to “figure 1” on page 5, paragraph 1, line 214. Figure S2 was changed into figure S1 in the supplement and page 3, paragraph 2 ,line 136.
Point 2: Why did Authors not consider to divide the patients into subgroups (with hypertension, dyslipidemia, etc.)?
Response 2: The analysis for each subgroup (e.g., hypertension, dyslipidemia, and diabetes mellitus) on primary and secondary outcomes was not different from the overall population. We added the outcome of each subgroup on page 5, paragraph 1, line 223-226 and table S2 (supplement).
Point 3: Did Authors consider sex as a variable that may cause the differences in the pharmacokinetic analysis?
Response 2: In this study, we considered sex as a variable that may explain the differences in the pharmacokinetic analysis. We used PrecisePK which includes sex as a one of the variable to calculate the pharmacokinetic analysis (page 3, paragraph 2, line 146-148). However, we did not find any differences in pharmacokinetic parameters between genders except vancomycin clearance (page 4, paragraph 3, line 184-187). The primary and secondary outcome in the male participant between two groups yield similar results with the overall population page 5, paragraph 3, line 223-226 and table S2 (supplement).
Point 4: Why did Authors did not take into consideration the protein plasma levels?
Response 4: We added protein plasma levels between two groups in table 1 (page 5) , in the result section on page 4, paragraph 3, line 176 and in the method on page 3, paragraph 1, line 115-116.
Reviewer 2 Report
The manuscript describes a pharmacist-led multidisciplinary team intervention to increase safety and efficacy of vancomycin therapy. The study does not provide new knowledge or methods (the Authors employed commercially available software and a widely used approach for vancomycin dose optimization), however, the obtained results may be of interest for clinical pharmacists in other countries, especially those where TDM is not popular. The study is well designed and the results are adequately reported and explained. However, several issues still may be improved:
It is not clear why the patients with advanced renal disease were excluded from analysis. The drug concentrations measured in these patients may be interesting for the readers and these patients need dosage adjustment more often than the patients with normal renal function. In addition, the software used involves extra drug factors option, such as “critically ill or ICU patient” or “patients on CRRT”.
The analytical method that was used for vancomycin determination in patient samples was not mentioned in the manuscript.
The information concerning initial vancomycin doses (in pre-PMT and PMT period) and the infusion length should be added to the Methods section.
It is not clear how the intervention influenced AUC value in the post-TMP period. The individual values of AUC (and concentrations) before and after TMP intervention should be shown.
Please provide PK parameters predicted by the software and AUC predicted and calculated after dosage adjustment.
Author Response
Response to Reviewer 2 Comments
Point 1: The manuscript describes a pharmacist-led multidisciplinary team intervention to increase safety and efficacy of vancomycin therapy. The study does not provide new knowledge or methods (the Authors employed commercially available software and a widely used approach for vancomycin dose optimization), however, the obtained results may be of interest for clinical pharmacists in other countries, especially those where TDM is not popular. The study is well designed and the results are adequately reported and explained. However, several issues still may be improved:
It is not clear why the patients with advanced renal disease were excluded from analysis. The drug concentrations measured in these patients may be interesting for the readers and these patients need dosage adjustment more often than the patients with normal renal function. In addition, the software used involves extra drug factors option, such as “critically ill or ICU patient” or “patients on CRRT”.
Response 1: In our study, because the schedules of patients on hemodialysis were unpredictable (time-wise) in the hospital and since most of the patients were not on CRRT, we decided to exclude this populations from our study. This is acknowledged as a limitation of our study on page 8, paragraph 2, line 345-346.
Point 2: The analytical method that was used for vancomycin determination in patient samples was not mentioned in the manuscript.
Response 2: We added the analytical method for vancomycin level determination in patient samples on page 3, paragraph 2, line 141- 143 and along with the reference (reference 20).
Point 3: The information concerning initial vancomycin doses (in pre-PMT and PMT period) and the infusion length should be added to the Methods section.
Response 3: We added information on initial vancomycin doses and the infusion length to the method on page 3, paragraph 2, line 138-141.
Point 4: Please provide PK parameters predicted by the software and AUC predicted and calculated after dosage adjustment.
Response 4: We added the PK parameters data predicted by the software on page 3, paragraph 1, line 116-118, in the result on page 4, paragraph 2, line 183-184,and Table 1 (page 5). We added the calculated and predicted AUC after dose adjustment on page 5, paragraph 1, line 214-215 and table 2 (page 6).
Reviewer 3 Report
This is study that demonstrates to be encouraged to intervation by the Pharmaist-led Multidisciplinary Team in patients, who used the vancomycin. A major point requiring revision for this article is the fact that the analysis is conducted on data from the patients that be administrated vancomycin (VCM). As outlined at the beginning of my review, there is certainly a need for increased evidence of effective and safe target concentrations. As it can be difficult to establish confirmed infection (MRSA infection) with organisms susceptible to VCM, reported patient numbers with such confirmed infections are low. Due to the difficulties described, it is likely that patients with infections susceptible to VCM are therefore missed and excluded from analyses. In that context, including all patients is of some benefit. However, the converse is then true, that some patients who do not have infections susceptible to VCM, are likely to be inadvertently included. This needs to be recognized as a limitation. Furthermore, in Table 1 it is presented that 13 and 21 patients in pre-PMT period and post-PMT period have bacteremia, respectively. These patients should be presented as a subgroup, with the respective VCM AUC levels for clinical cure and failure; the remaining "unknown origin" should also be presented as an additional subgroup, with corresponding VCM AUC data. Whereas the data numbers in subgroups are too low for a strong interpretation, the mean/median troughs could at least be compared to ascertain whether there were any differences in these 2 subgroups. Any information on local MIC data for the organisms detailed would also support further interpretation.
Author Response
Response to Reviewer 3 Comments
Point 1: This is study that demonstrates to be encouraged to intervention by the Pharmacist-led Multidisciplinary Team in patients, who used the vancomycin. A major point requiring revision for this article is the fact that the analysis is conducted on data from the patients that be administrated vancomycin (VCM). As outlined at the beginning of my review, there is certainly a need for increased evidence of effective and safe target concentrations. As it can be difficult to establish confirmed infection (MRSA infection) with organisms susceptible to VCM, reported patient numbers with such confirmed infections are low. Due to the difficulties described, it is likely that patients with infections susceptible to VCM are therefore missed and excluded from analyses. In that context, including all patients is of some benefit. However, the converse is then true, that some patients who do not have infections susceptible to VCM, are likely to be inadvertently included. This needs to be recognized as a limitation.
Response 1: We acknowledged these points in our limitation page 8, paragraph 1, line 351-355.
Point 2: Furthermore, in Table 1 it is presented that 13 and 21 patients in pre-PMT period and post-PMT period have bacteremia, respectively. These patients should be presented as a subgroup, with the respective VCM AUC levels for clinical cure and failure; the remaining "unknown origin" should also be presented as an additional subgroup, with corresponding VCM AUC data. Whereas the data numbers in subgroups are too low for a strong interpretation, the mean/median troughs could at least be compared to ascertain whether there were any differences in these 2 subgroups.
Response 2: Because the sample size is small and there are some complication in formatting the table 2, we decided to add the text to describe those detail on page 5, paragraph 1, line 223-226 and table S2 (supplement)
Point 3: Any information on local MIC data for the organisms detailed would also support further interpretation.
Response 3: We added information on local MIC data as suggested on page 4, paragraph 2, line 182-183.
Round 2
Reviewer 2 Report
The manuscript has been significantly improved.
I have one minor comment: in Table 1 – t0.5 (1 h) in post-PMT period is probably not correct.
Author Response
Point 1: The manuscript has been significantly improved. I have one minor comment: in Table 1 – t0.5 (1 h) in post-PMT period is probably not correct.
Response 1: Thank you. We correct the half-life in post-PMT period in from 1 ± 6.5 to 10 ± 6.5 on page 5, table 1.
Reviewer 3 Report
no comment
Author Response
Point 1: no comment.
Response 1: Thank you.
